# Longitudinal Study on the Effect of Onboard Service on Seafarers’ Health Statuses

**DOI:** 10.3390/ijerph20054497

**Published:** 2023-03-03

**Authors:** Andrea Russo, Rosanda Mulić, Ivana Kolčić, Matko Maleš, Iris Jerončić Tomić, Luka Pezelj

**Affiliations:** 1Faculty of Maritime Studies, University of Split, Ruđera Boškovića 37, 21000 Split, Croatia; 2School of Medicine, University of Split, Šoltanska 2, 21000 Split, Croatia

**Keywords:** body mass index (BMI), bioelectrical impedance (BIA), body composition, seaman, maritime health, occupational health, aging, metabolic syndrome, chronic health effects, health risk

## Abstract

Seafaring is considered one of the most stressful professions. Stressors in seafaring lead to typical symptoms of stress, such as insomnia, loss of concentration, anxiety, lower tolerance of frustration, changes in eating habits, psychosomatic symptoms and diseases, and overall reduced productivity, with the possibility of burnout and chronic responsibility syndrome. It has been previously determined that seafarers belong to high-risk occupations in terms of developing metabolic syndrome, and according to their BMIs, almost 50% of all seafarers belong to the overweight and obesity categories. This is the first longitudinal study conducted with the aim of using the BIA method to determine the anthropometrical changes that occur during several weeks of continuous onboard service. This study included an observed group consisting of 63 professional seafarers with 8 to 12 weeks of continuous onboard service and a control group of 36 respondents from unrelated occupations. It was determined that Croatian seafarers fit into the current world trends regarding overweight and obesity among the seafaring population, with the following percentages in the BMI categories: underweight, 0%; normal weight, 42.86%; overweight, 39.68%; and obesity, 17.46%. It was established that the anthropometric statuses of the seafarers significantly changed during several weeks of continuous onboard service. Seafarers who served on board for 11 weeks lost 0.41 kg of muscle mass, whereas their total fat mass increased by 1.93 kg. Changes in anthropometric parameters could indicate deterioration of seafarers’ health statuses.

## 1. Introduction

The atypicality and specificities of work and family life, i.e., social life, are the main characteristics and differences of seafarers’ lives in comparison to those of the rest of the working population [1]. The variety and speed of environmental changes and exposure to continuous noise and vibration entailed make it hard to maintain psychophysical homeostasis, not to mention other stressors that arise from the specifics of the maritime profession, which are still insufficiently taken into account [2]. At least one environmental factor, such as excessively cold or warm ambient temperatures, odors, noise, poor bedding conditions, or ambient light during sleeping in cabins disturbs 91.6% of seafarers [3].

Stressors in seafaring lead to typical symptoms of stress, such as insomnia, loss of concentration, anxiety, lower tolerance of frustration, changes in eating habits, psychosomatic symptoms and diseases, and overall reduced productivity, with the possibility of burnout and chronic responsibility syndrome [4].

With circadian work rhythms such as the 6:6 and 4:8 shift systems, body recovery and sleep are interrupted and often insufficient. During night work on board in the 6:6 (midnight to 6:00 a.m.) and 4:8 (midnight to 4:00 a.m.) systems, seafarers experience increased sleepiness with shorter sleep episodes [4].

Metabolic health, cancer risk, cardiovascular health, and mental health are further compromised by shift work, especially night work. This is due to the problems caused by the shift-work lifestyle, which are mainly manifested in chronic sleep deprivation, sympathovagal and hormonal imbalance, inflammation, impaired glucose metabolism, and unregulated cell cycles. As a result, such long-term conditions lead to a number of health disorders such as obesity, metabolic syndrome, type II diabetes, gastrointestinal dysfunction, impaired immune function, cardiovascular disease, excessive sleepiness, mood and social disorders, and increased risk of cancer [5].

Compared with that of other transportation sectors, fatigue in the maritime sector has been much less researched. Fixed and rotating work schedules, along with cultural and commercial pressures, directly affect seafarers’ physical and mental health [4,6].

With knowledge that during their service on board, seafarers have a limited influence on quality and quantity of food [7], and that nutritional problems are even more pronounced in multiethnic crews with different eating habits, it is clear that the physical and psychological conditions of seafarers may imperceptibly deteriorate [4].

An individual’s ability to adequately cope with the demands of such a maritime occupation depends on that individual’s state of physical and mental health. An extremely demanding maritime occupation, which limits a person in maintaining the usual way of life on land in terms of food choices, regular sleep, and often the inability to exercise, can lead to a gradual loss of physical and mental fitness, which can ultimately lead to human error, illness, and disabilities related to seafarers’ work [8].

A diet that does not include enough fresh fruits and vegetables can contribute to fatigue and has an overall negative impact on seafarers’ health [9,10]. In addition, the circadian rhythm of work affects digestion, which is most productive during the day and much less so at night, even when a person is awake and in a working rhythm [11]. Gastrointestinal disorders are very common in people who eat outside of traditional mealtimes and tend to worsen with consumption of tea, coffee, alcohol, nicotine, and some medications and supplements. Night workers are five times more likely to contract peptic ulcers than are day workers [12].

Exercise and good physical fitness have beneficial effects on the body and psyche, help in coping with stress, and can help reduce a person’s susceptibility to certain diseases and infections [13]. Some of the anthropometric methods for assessing a person’s health status are analysis of body composition and evaluation of body structure. The most-used methods are bioelectrical impedance (BIA) and the body mass index (BMI) [14]. The BMI is widely accepted and used as a standard test, and BIA is a valid and precise method for determining the body compositions of normal, healthy people [15] and athletes [16]. Due to fast and noninvasive measurement, BIA is widely used within the athlete population, but it has never been used in the population of professional seafarers.

Therefore, the aim of this study was to determine the body compositions of Croatian seafarers and investigate changes in anthropometric parameters during continuous onboard service.

## 2. Materials and Methods

### 2.1. Subject and Variable Sample

The subject sample included 99 adults from Croatia (Caucasian), divided into a control group and an experimental group. The control group included 36 subjects with a mean chronological age of 33.56 ± 8.49 years, a mean body height of 183.22 ± 5.58 cm, and a mean body mass of 93.15 ± 15.36 kg. The sample in the control group was a convenience sample selected to resemble the experimental group in the initial testing. Furthermore, the test subjects in the control group were selected on the condition that they did not perform jobs that may be similar to those of the seafarers, or that involve long-term absence from home (e.g., drivers, pilots, soldiers, coaches, athletes, etc.). The experimental group included 63 subjects with a mean chronological age of 35.00 ± 8.08 years, a mean body height of 183.73 ± 5.94 cm, and a mean body mass of 89.43 ± 10.82 kg. The subject sample included professional seafarers who serve on merchant ships as officers aboard various types of ships and for various companies. To make the experimental group as homogenous as possible, subjects whose service aboard was shorter than 8 weeks or longer than 12 weeks were excluded from the sample. This period did not include the “idle” time between testing and departure, i.e., return from the ship. All subjects were measured on two occasions, the initial and final measurements, performed during morning hours. The subjects in the experimental group (professional seafarers) were measured within seven days before departure and within seven days after returning home. Subjects in the control group were measured with random selection in the final testing, 8 to 12 weeks after the initial testing.

Two anthropometric variables were measured, body height and body mass, which were then used to calculate the body mass index. All measurements were taken according to the International Society for the Advancement of Kinanthropometry—ISAK protocol [17]. Furthermore, the subjects were measured with a Tanita BC-418 (Tanita Corp., Tokyo, Japan) device following the recommendations of Kyle et al. [18], and the results of the following anthropometric measures were determined using the bioelectrical impedance method: the body fat percentage, fat mass, visceral fat, metabolic age, fat-free mass, total body water, extracellular water, intracellular water, muscle mass, the skeletal muscle index, bone mass, and the basal metabolic rate.

### 2.2. Description of Body Composition Measures

Body composition measures (the body fat percentage, fat mass, visceral fat, metabolic age, fat-free mass, total body water, extracellular water, intracellular water, muscle mass, the skeletal muscle index, bone mass, and the basal metabolic rate) were determined with the bioelectrical impedance method, using a Tanita BC-418 device (Tanita Corp., Tokyo, Japan). The subjects were measured barefoot and in dry underwear. The “body type” setting was set to “normal” for all subjects, whereas the “weight of clothes” was set to 0.0 kg.

Body Fat Percentage—the proportion of fat to the total body weight.

Fat Mass—the actual weight of the fat in the body.

Visceral fat—fat located deep in the core abdominal area, surrounding and protecting the vital organs.

Muscle Mass—the predicted weight of muscle in the body.

Total Body Water—the total amount of fluid in the body, expressed as a percentage of the total weight.

Extracellular Water—body fluid found outside of cells.

Intracellular Water—fluid found inside cells.

Bone Mass—the predicted weight of bone mineral in the body.

Basal Metabolic Rate—the daily minimum level of energy or calories the body requires when at rest (including sleeping) in order to function effectively.

Metabolic Age—a comparison of the basal metabolic rate (BMR) to the BMR average of a chronological age group. If the metabolic age is higher than the actual age, it is an indication that improving the metabolic rate is needed.

Skeletal Muscle Index—the ratio of the muscle in the arms and legs to height.

### 2.3. Methods of Data Analysis

For all the measured variables and for each subject sample separately, the following descriptive parameters were calculated: arithmetic mean (AM); standard deviation (SD); median (M), minimum (MIN) and maximum (MAX) results; and the coefficients of asymmetry (SKEW) and peakedness (KURT) of result distribution. Normality of distribution was tested with the Kolmogorov–Smirnov test (KS). The differences in initial testing in chronological age, anthropometric characteristics, and body composition measures between the control and experimental groups were determined using the independent samples *t*-test. The differences between the initial and final measurements of chronological age, anthropometric characteristics, and body composition measures between the control and experimental groups were determined using the *t*-test for dependent samples.

For each measured variable, the differences between the initial and final tests of the control and experimental groups were calculated and arithmetic means were determined. The differences between the initial and final tests of chronological age, anthropometric characteristics, and body composition measures in the control and experimental groups were determined using the independent samples *t*-test. The data were analyzed using Statistica Ver 11.0 (SoftStat, SAD, Tulsa, OK, USA).

## 3. Results

Table 1 presents the results of the Kolmogorov–Smirnov test of anthropometric variables indicate that no variable exceeded the cutoff value of the test, which was 0.23 for the observed sample. This indicates that there were no significant deviations of the variables from normal distribution, and all variables were suitable for further parametric statistical analysis.

Table 2 presents the results of the Kolmogorov–Smirnov test of anthropometric variables indicate that no variable exceeded the cutoff value of the test, which was 0.17 for the observed sample. This indicates that there were no significant deviations of the variables from normal distribution, and all variables were suitable for further parametric statistical analysis.

Table 3 presents that in the initial *t*-test measurement, no significant differences were found between the control and experimental groups in the arithmetic mean scores of the measured variables.

Table 4 presents that the *t*-test revealed significant differences between the initial and final measurements in the experimental group for the following variables: age, weight, the body mass index, the fat percentage, fat mass, visceral fat, metabolic age, fat-free mass, total body water, intracellular water, and muscle mass.

Table 5 presents that the t-test revealed significant differences between the control and experimental groups in the changes in the values of the measured variables between the initial and final measurements in the following variables: weight, the body mass index, fat percentage, fat mass, visceral fat, and metabolic age.

## 4. Discussion

The BMI was the most frequently measured/analyzed anthropometric variable in previous research on a sample of professional seafarers [19,20,21,22,23,24,25]. In this study, the following proportions of professional seafarers regarding the BMI categories to which they belong were determined: underweight, 0%; normal weight, 42.86%; overweight, 39.68%; and obesity, 17.46%. We should compare the obtained results with those of other authors with great caution because the BMI depends, among other things, on the cultural and ethnic characteristics of the population [26]. In a sample of 1155 subjects, Nittari [24] found an average BMI of 25.7 kg/m^2^, and the proportions were very similar to those found in this study: underweight, 0.8%; normal weight, 47.20%; overweight, 40.80%; and obesity, 11.20%. Similar results were found in a study conducted by Gamo Sagaro in 2021 [25], in which the mean BMI was 25.55 kg/m^2^ and the following percentages were determined in the BMI categories: underweight, 0%; normal weight, 51.90%; overweight, 39.30%; and obesity, 8.50%. The comparison with the 2021 study is even more significant because the average age of the subjects (N = 603) was 37.37 years: very similar to the sample in this study. The higher proportion of obesity and higher mean BMI values in these seafarers compared to the Nittari research can be explained with the fact that 51% of subjects in that study were Filipinos and Indians, who by default have a lower tendency to be overweight and obese [27]. Results almost identical to the results of this study were determined in Hoeyer’s 2005 [19] study on seafarers aged 25–44 years (N = 613): underweight, 2.8%; normal weight, 40.0%; overweight, 38.8%; and obesity, 18.4%.

A higher proportion of overweight and obese seafarers compared to the observed sample was determined in a study conducted by Hansen in 2011 [20], which, among other things, indicated an increase in the frequency of overweight among seafarers. We can conclude that Croatian seafarers fit into the current world trends regarding overweight and obesity among the seafaring population, which is defined as one of the main health problems of today. However, in comparison of the BMIs of Croatian seafarers with WHO data for the Croatian general population, Croatian seafarers have lower mean BMI values and thus a lower proportion of overweight and obesity. The control group also had lower mean BMI values than the general population, according to the WHO [27].

BIA is a very fast, simple, and reliable method for body composition analysis [28,29,30,31]. Although BIA measurement is widely used among top athletes [32,33], it has not been used in the population of professional seafarers until now. Moreover, it has not even been used in the population of drivers, who, along with seafarers, belong to the group of highest-risk workers [34]. The observed sample of seafarers and the control group had lower %BF values than did maritime university students [22], even though the subjects in this study were significantly older and the percentage of fat tissue has a tendency to increase with age [35]. This can also be explained with the fact that the studies did not use body-composition-analysis instruments from the same manufacturer. In a study on a sample of professional firefighters, [36] the same analysis equipment was used as in this study, and the results indicated similar %BF values as in seafarers of the same age, i.e., slightly higher %BF values in older firefighters, as expected. In addition to determining anthropometrical characteristics of seafarers, this study aimed to analyze changes in body compositions of seafarers during service on board. To the authors’ knowledge, this is the first longitudinal study on the population of professional seafarers. 

To ensure an unambiguous interpretation of results, this study also included a control group of subjects, which did not significantly differ statistically from the experimental group. During 10.97 weeks of onboard service, the total body mass of the professional seafarers increased by 1.50 kg. Although the change in total body mass compared to that of the control group was significant, it should not be a cause for concern in real life. However, analysis of body composition revealed fundamental problems that, at first glance, remained hidden in the relatively small change in total body mass. During their service on board, the seafarers on average, lost 0.41 kg of their total muscle mass, whereas their total fat masses increased by 1.93 kg. Of course, this “negative” transformation was also reflected in other indicators of body composition. Thus, an increase of 1.81 percentage points in the percentage of body fat and an increase of 0.73 in the visceral fat rating were determined.

Average, muscle mass loss of 0.41 kg and total body fat increase of 1.93 kg was recorded among the sample of subjects who served on board for 11 weeks. An increased proportion of fat mass in the body structure results in risk of metabolic syndrome, which is characterized with visceral obesity associated with insulin resistance, arterial hypertension, dyslipidemia, diabetes, and glucose intolerance. Possible causes of these rapid anthropometric changes are physical inactivity on board and circadian rhythm disorders with sleep disorders. Lack of sleep and circadian sleep disorders are symptoms of many conditions. Jepsen concluded that lack of sleep is associated with obesity [37], and it is debated whether circadian sleep disorders are the causes or the consequences of some neurodegenerative diseases [38,39].

Body composition is a much better indicator of the degree of nutritional status than the body mass index is because obesity is not defined as increased body mass but as an increased proportion of adipose tissue in body mass. Among the study subjects, an average increase of 1.81 percentage points % in the percentage of body fat and an average increase of 0.73 in the visceral fat rating was determined.

## 5. Conclusions

In this paper, anthropometrical characteristics of professional seafarers, which can certainly be a point of reference for future research, were determined. Furthermore, this is one of the rare studies in which the problem of the influence of onboard service on professional seafarers’ health was approached through a longitudinal study. It was established that the anthropometric statuses of the seafarers significantly changed during several weeks of continuous onboard service. These changes in anthropometric parameters could indicate deterioration of seafarers’ health statuses. We can only speculate about the causes of those anthropometric changes in a relatively short interval. The main shortcomings of this study are reflected in the fact that no external factors were measured. Therefore, it is recommended for future studies to include tests and methods aimed at detecting possible negative factors such as diet, sleep, psychological stress, etc. It is also recommended to repeat this research on seafarers who perform other types of work on merchant fleets (engineers, auxiliary staff, etc.), seafarers of other maritime occupations (fishermen, skippers, etc.), and other professionals who must leave home for a long time to perform their jobs (pilots, drivers, seasonal workers, etc.).

## Figures and Tables

**Table 1 ijerph-20-04497-t001:** Descriptive parameters and sensitivity of the control group (N = 36).

Variable	AM ± SD	M	MIN	MAX	SKEW	KURT	KS
Initial Testing Age (Years)	33.56 ± 8.49	34.56	18.77	52.15	0.20	−0.78	0.10
Height (cm)	183.22 ± 5.58	182.00	170.00	196.00	0.12	−0.02	0.13
Body Mass (kg)	93.15 ± 15.36	93.90	61.30	118.80	−0.19	−0.76	0.09
Body Mass Index (kg/m^2^)	27.76 ± 4.58	27.80	20.00	35.20	−0.02	−1.38	0.15
Body Fat Percentage (%)	21.05 ± 7.68	22.05	5.60	33.70	−0.25	−1.23	0.14
Fat Mass (kg)	20.57 ± 9.93	21.00	3.40	37.50	0.05	−1.36	0.14
Visceral Fat (Rating)	7.69 ± 4.96	8.00	1.00	17.00	0.00	−1.34	0.13
Metabolic Age (Years)	35.31 ± 17.93	41.00	12.00	63.00	−0.17	−1.63	0.18
Fat-Free Mass (kg)	72.58 ± 7.34	71.90	56.40	84.70	−0.30	−0.22	0.10
Total Body Water (%)	51.84 ± 5.15	51.15	41.60	61.30	0.01	−0.66	0.13
Extracellular Water (%)	20.62 ± 2.02	20.45	16.50	24.00	−0.23	−0.60	0.10
Intracellular Water (%)	31.22 ± 3.39	30.90	24.80	38.80	0.32	−0.40	0.09
Muscle Mass (kg)	69.00 ± 6.70	68.35	53.60	80.60	−0.30	−0.22	0.09
Skeletal Muscle Index	9.57 ± 0.89	9.64	7.84	11.23	−0.18	−0.94	0.15
Bone Mass (kg)	3.57 ± 0.34	3.55	2.80	4.10	−0.25	−0.23	0.13
Basal Metabolic Rate (kcal)	2151.03 ± 234.09	2122.50	1668.00	2515.00	−0.17	−0.51	0.12
Final Testing Age (years)	33.76 ± 8.49	34.75	18.98	52.30	0.20	−0.78	0.10
KS test = 0.23							

Legend: N—number of subjects, AM—arithmetic mean, SD—standard deviation, M—median, MIN—minimum result, MAX—maximum result, SKEW—coefficient of asymmetry of result distribution, KURT—coefficient of peakedness of result distribution, KS—result of the Kolmogorov–Smirnov test, KS MAXD—cutoff value of the Kolmogorov–Smirnov test.

**Table 2 ijerph-20-04497-t002:** Descriptive parameters and sensitivity of the experimental group (N = 63).

Variable	AM ± SD	M	MIN	MAX	SKEW	KURT	KS
Initial Testing Age (Years)	35.00 ± 8.08	33.63	23.18	52.28	0.41	−1.02	0.12
Height (cm)	183.73 ± 5.94	183.00	169.00	196.00	−0.18	−0.36	0.13
Body Mass (kg)	89.43 ± 10.82	90.40	61.30	112.40	−0.22	−0.37	0.10
Body Mass Index (kg/m^2^)	26.51 ± 3.20	26.70	20.00	34.60	0.27	−0.55	0.13
Body Fat Percentage (%)	19.23 ± 5.29	19.00	5.60	29.60	−0.09	−0.52	0.07
Fat Mass (kg)	17.58 ± 6.33	16.30	3.40	30.90	0.24	−0.62	0.09
Visceral Fat (Rating)	6.60 ± 3.44	6.00	1.00	13.00	0.26	−0.97	0.11
Metabolic Age (Years)	31.65 ± 13.88	31.00	12.00	56.00	0.26	−1.12	0.11
Fat-Free Mass (kg)	71.84 ± 6.63	71.20	57.90	84.70	0.12	−0.49	0.08
Total Body Water (%)	51.20 ± 4.99	50.60	42.50	61.30	0.28	−0.62	0.09
Extracellular Water (%)	20.20 ± 1.60	20.30	16.50	23.60	−0.05	−0.52	0.08
Intracellular Water (%)	31.01 ± 3.59	30.60	25.30	38.80	0.39	−0.59	0.07
Muscle Mass (kg)	68.30 ± 6.33	67.70	55.00	80.60	0.12	−0.49	0.08
Skeletal Muscle Index	9.31 ± 0.83	9.21	7.84	11.33	0.10	−1.08	0.14
Bone Mass (kg)	3.55 ± 0.30	3.50	2.90	4.10	0.09	−0.56	0.11
Basal Metabolic Rate (kcal)	2112.37 ± 210.75	2099.00	1668.00	2507.00	0.11	−0.56	0.08
Final Testing Age (Years)	35.21 ± 8.09	33.90	23.33	52.51	0.41	−1.02	0.12
KS test = 0.17							

Legend: N—number of subjects, AM—arithmetic mean, SD—standard deviation, M—median, MIN—minimum result, MAX—maximum result, SKEW—coefficient of asymmetry of result distribution, KURT—coefficient of peakedness of result distribution, KS—result of the Kolmogorov–Smirnov test, KS MAXD—cutoff value of the Kolmogorov–Smirnov test.

**Table 3 ijerph-20-04497-t003:** Statistical significance of differences in arithmetic means (*t*-test) of chronological age and anthropometric characteristics of the control and experimental groups in the initial measurement.

Variable	CON (N = 36)	EXP (N = 63)	t	*p*
	AM ± SD	AM ± SD		
Initial Testing Age (Years)	33.56 ± 8.49	35.00 ± 8.08	−0.84	0.404
Height (cm)	183.22 ± 5.58	183.73 ± 5.94	−0.42	0.677
Body Mass (kg)	93.15 ± 15.36	89.43 ± 10.82	1.41	0.162
Body Mass Index (kg/m^2^)	27.76 ± 4.58	26.51 ± 3.20	1.60	0.114
Body Fat Percentage (%)	21.05 ± 7.68	19.23 ± 5.29	1.39	0.168
Fat Mass (kg)	20.57 ± 9.93	17.58 ± 6.33	1.83	0.070
Visceral Fat (Rating)	7.69 ± 4.96	6.60 ± 3.44	1.29	0.200
Metabolic Age (Years)	35.31 ± 17.93	31.65 ± 13.88	1.13	0.261
Fat-Free Mass (kg)	72.58 ± 7.34	71.84 ± 6.63	0.51	0.613
Total Body Water (%)	51.84 ± 5.15	51.20 ± 4.99	0.60	0.550
Extracellular Water (%)	20.62 ± 2.02	20.20 ± 1.60	1.14	0.255
Intracellular Water (%)	31.22 ± 3.39	31.01 ± 3.59	0.29	0.775
Muscle Mass (kg)	69.00 ± 6.70	68.30 ± 6.33	0.51	0.609
Skeletal Muscle Index	9.57 ± 0.89	9.31 ± 0.83	1.51	0.134
Bone Mass (kg)	3.57 ± 0.34	3.55 ± 0.30	0.37	0.711
Basal Metabolic Rate (kcal)	2151.03 ± 234.09	2112.37 ± 210.75	0.84	0.401
Final Testing Age (Years)	33.76 ± 8.49	35.21 ± 8.09	−0.84	0.403

Legend: N—number of subjects, AM—arithmetic mean, SD—standard deviation, t—ratio between the differences of two arithmetic means and their standard errors, *p*—level of statistical significance.

**Table 4 ijerph-20-04497-t004:** Statistical significance of differences in arithmetic means (*t*-test) of chronological age and anthropometric characteristics of the initial and final tests of the experimental group.

Variable	EXPERIMENTAL GROUP (N = 63)		t	*p*
	INITAL	FINAL		
	AM ± SD	AM ± SD		
Age (Years)	35.00 ± 8.08	35.21 ± 8.09	−45.04	0.000 ***
Height (cm)	183.73 ± 5.94	183.73 ± 5.94		
Weight (kg)	89.43 ± 10.82	90.93 ± 10.98	−5.26	0.000 ***
Body Mass Index (kg/m^2^)	26.51 ± 3.20	26.96 ± 3.25	−5.30	0.000 ***
Body Fat Percentage (%)	19.23 ± 5.29	21.04 ± 5.16	−9.62	0.000 ***
Fat Mass (kg)	17.58 ± 6.33	19.52 ± 6.46	−9.08	0.000 ***
Visceral Fat (Rating)	6.60 ± 3.44	7.33 ± 3.53	−8.56	0.000 ***
Metabolic Age (Years)	31.65 ± 13.88	36.16 ± 14.27	−10.77	0.000 ***
Fat-Free Mass (kg)	71.84 ± 6.63	71.41 ± 6.49	2.27	0.027 *
Total Body Water (%)	51.20 ± 4.99	50.62 ± 4.88	3.39	0.001 ***
Extracellular Water (%)	20.20 ± 1.60	20.23 ± 1.60	−0.65	0.52
Intracellular Water (%)	31.01 ± 3.59	30.39 ± 3.47	4.34	0.000 ***
Muscle Mass (kg)	68.30 ± 6.33	67.89 ± 6.19	2.27	0.027 *
Skeletal Muscle Index	9.31 ± 0.83	9.26 ± 0.84	1.63	0.11
Bone Mass (kg)	3.55 ± 0.30	3.53 ± 0.30	1.99	0.051
Basal Metabolic Rate (kcal)	2112.37 ± 210.75	2105.90 ± 206.95	1.09	0.28

Legend: N—number of subjects, AM—arithmetic mean, SD—standard deviation, t—ratio between the differences of two arithmetic means and their standard errors, *p*—level of statistical significance, *—significant difference at the level of *p* ≤ 0.05, ***—significant difference at the level of *p* ≤ 0.005.

**Table 5 ijerph-20-04497-t005:** Statistical significance of differences in arithmetic means (*t*-test) of chronological age and anthropometric characteristics in the changes in values of the measured variables between the initial and final measurements in the control and experimental groups.

Variable	CON (N = 36)	EXP (N = 63)	t	*p*
	AM ± SD	AM ± SD		
Initial Testing Age (Years)	0.21 ± 0.04	0.21 ± 0.04	0.35	0.731
Height (cm)	0.00 ± 0.00	0.00 ± 0.00		
Weight (kg)	−0.54 ± 2.13	1.50 ± 2.27	4.40	0.000 ***
Body Mass Index (kg/m^2^)	−0.17 ± 0.62	0.45 ± 0.67	4.49	0.000 ***
Body Fat Percentage (%)	−0.22 ± 1.29	1.81 ± 1.49	6.83	0.000 ***
Fat Mass (kg)	−0.41 ± 1.48	1.93 ± 1.69	6.93	0.000 ***
Visceral Fat (Rating)	−0.25 ± 0.60	0.73 ± 0.68	7.20	0.000 ***
Metabolic Age (Years)	0.14 ± 0.96	4.51 ± 3.32	7.70	0.000 ***
Fat-Free Mass (kg)	−0.13 ± 1.62	−0.43 ± 1.52	−0.93	0.353
Total Body Water (%)	−0.08 ± 1.45	−0.58 ± 1.36	−1.74	0.085
Extracellular Water (%)	−0.07 ± 0.41	0.03 ± 0.43	1.19	0.236
Intracellular Water (%)	−0.01 ± 1.08	−0.61 ± 1.12	−2.63	0.010
Muscle Mass (kg)	−0.13 ± 1.54	−0.41 ± 1.44	−0.91	0.366
Skeletal Muscle Index	−0.02 ± 0.27	−0.05 ± 0.25	−0.49	0.626
Bone Mass (kg)	0.00 ± 0.09	−0.02 ± 0.09	−1.19	0.235
Basal Metabolic Rate (kcal)	−6.72 ± 49.41	−6.46 ± 46.88	0.03	0.979

Legend: N—number of subjects, AM—arithmetic mean, SD—standard deviation, t—ratio between the differences of two arithmetic means and their standard errors, *p*—level of statistical significance, ***—significant difference at the level of *p* ≤ 0.005.

## Data Availability

The data presented in this study are available on request from the corresponding authors.

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
