# Peer review of "Longitudinal Study on the Effect of Onboard Service on Seafarers’ Health Statuses"

_ijerph, 2023, doi:10.3390/ijerph20054497_

Round 1

Reviewer 1 Report

I have had the opportunity to review a manuscript entitled “Longitudinal Study on the Effect of Onboard Service on Seafarers' Health Status”. The authors investigated anthropometrical changes among seafarers after 8 to 12 weeks of continuous onboard service in comparison to the control group. Although the subject of the study is interesting, some reforms are needed to make the study clear.

Abstract

The authors should provide more information about the methods and the main findings. Please describe the background briefly and include a conclusion to the abstract.

Introduction

It is difficult to follow the introduction. The authors should consider the flow and integrity of the introduction. Ideally, three or four paragraphs with the main concepts as the following can be constructive: Seafaring importance and exposures, the effects of these exposures on the seafarer’s health, a review of the previous studies among seafarers related to this study, the novelty and the aims of this study.   

Methods 

Please provide information about how the study participants were selected. 

Why do the authors consider 8 to 12 weeks as an exposure period? Is the selection of this period related to the number of participants beyond these limits?

Some inclusion criteria were reported for the unexposed group but no information (job title) was provided about the final subjects included in the unexposed group. Is the unexposed group can be exposed to the same agents (noise, heat, shift work, etc.) as the seafarers?

In my opinion, given that the occupational exposure agents were not measured, it doesn’t make sense to compare the changes in the BIA variables in the seafarers and the unexposed group. Investigating the variations in the BIA parameters among seafarers between the initial and final measurements (Paired t-test) can be more constructive.

Another important issue is related to the length of rest time for seafarers. Is the initial measurement can be affected by previous exposure time?        

Please remove duplicate information (Sections 2.2 and 2.3 can be combined).   

Results

The table 1-2 can be combined into a single table (only the mean and SD parameters may be sufficient.)

Discussion

Please provide a limitations section to the discussion. For example, one of the main shortcomings of this study is the confounding effects.

Subdividing the discussion into different sections with a specific main concept (for example, the different sections for BMI, BIA, and advantages and limitations) can be useful.

Author Response

Response to Reviewer 1 Comments

Point 1: The authors should provide more information about the methods and the main findings. Please describe the background briefly and include a conclusion to the abstract.

Response 1: Added

Point 2: It is difficult to follow the introduction. The authors should consider the flow and integrity of the introduction. Ideally, three or four paragraphs with the main concepts as the following can be constructive: Seafaring importance and exposures, the effects of these exposures on the seafarer’s health, a review of the previous studies among seafarers related to this study, the novelty and the aims of this study.   

Response 2: Wa added and exchanged some pharagraphs of introduction. Hopefuly it will make the introduciton more clear.

In Discussion chapter is “all availibele scientific literature” which include any of anthropometric paraemeters of seafarrers. So it is kind of “review paper” within our scientific paper so maybe there is no need for including the “previous studies” in the itroduction. 

Point 3: Please provide information about how the study participants were selected. 

Why do the authors consider 8 to 12 weeks as an exposure period? Is the selection of this period related to the number of participants beyond these limits?

Response 3: Most common seafaring contract for onboard services are from 1,5 months to 3 months period which relates with 8-12 weeks period. We included bigger sample of seafarers in the initial testing but their onboard service was disturbed with wast issues. Few of them had to cancel contrat shortly after coming on-board due to the private issues (after few weeks), and few of them combied double on-board period so they were on board for a time longer then 5 months. All other sample fited in the “normal” seafaring contract range.

Point 4: Some inclusion criteria were reported for the unexposed group but no information (job title) was provided about the final subjects included in the unexposed group. Is the unexposed group can be exposed to the same agents (noise, heat, shift work, etc.) as the seafarers?

Response 4: Criteria for unexposed group was to exclude similar “long-term absence from home” jobs. So theoretcaly the unexposed group could be exposed by some of the agents (noise, heat, shift work, bad diet, ect.) but not for an extended non-stop 8-12 weeks period.  

Point 5: In my opinion, given that the occupational exposure agents were not measured, it doesn’t make sense to compare the changes in the BIA variables in the seafarers and the unexposed group.

Response 5: There is insuficient amount of scientific ressearch in the field of seafaring, and this research is the first longitudinal study in the field. In this research we dermined that seafarers deteriorated the body composition during the onboard service. We are aware of the limitations of the study as we didn’t include possible negative factors which could couse this deterioration (diet, sleep, psychological stress), but findings of this research are good fundament for the future research which should include outside factors.

Point 6: Investigating the variations in the BIA parameters among seafarers between the initial and final measurements (Paired t-test) can be more constructive.

Response 6: We added the table of the initaial and final test of experimental gorup.

 Table 4. Statistical significance of differences in arithmetic means (T-test) of chronological age and anthropometric characteristics of the initial and final test of experimental group.

Variables

EXPERIMENTAL GROUP (N=63)

t

p=

INITAL

FINAL

AM±SD

AM±SD

Age (years)

35.00±8.08

35.21±8.09

-45.04

0.000***

Height (cm)

183.73±5.94

183.73±5.94

Weight (kg)

89.43±10.82

90.93±10.98

-5.26

0.000***

Body Mass Index (kg/m2)

26.51±3.20

26.96±3.25

-5.30

0.000***

Body Fat Percentage (%)

19.23±5.29

21,04±5.16

-9.62

0.000***

Fat Mass (kg)

17.58±6.33

19.52±6.46

-9.08

0.000***

Visceral Fat (rating)

6.60±3.44

7.33±3.53

-8.56

0.000***

Metabolic Age (years)

31.65±13.88

36.16±14.27

-10.77

0.000***

Fat-Free Mass (kg)

71.84±6.63

71.41±6.49

2.27

0.027*

Total Body Water (%)

51.20±4.99

50.62±4.88

3.39

0,001***

Extracellular Water (%)

20.20±1.60

20.23±1.60

-0.65

0.52

Intracellular Water (%)

31.01±3.59

30.39±3.47

4.34

0.000***

Muscle Mass (kg)

68.30±6.33

67.89±6.19

2.27

0,027*

Skeletal Muscle Index

9.31±0.83

9.26±0.84

1.63

0.11

Bone Mass (kg)

3.55±0.30

3.53±0.30

1.99

0.051

Basal Metabolic Rate (kcal)

2112.37±210.75

2105.90±206.95

1.09

0.28

Point 7: Another important issue is related to the length of rest time for seafarers. Is the initial measurement can be affected by previous exposure time?     

Response 7: Initial measurement can not be affected by previus exposure time, as we are measuring the difference between initial and final period. Concluision is in whatewer phisical “shape” you come on board, living and working on boat will make you worst.

Point 8: Please remove duplicate information (Sections 2.2 and 2.3 can be combined).   

Response 8: Corected

Point 9: The table 1-2 can be combined into a single table (only the mean and SD parameters may be sufficient.)

Response 9: As there is no sufficient literature in the field od seafaring, and this is first research of this type in a field, we would like to keep the subject description parameters (mean, min., max., skew, kurt) as they could be very valuable for the researchers in the the future subject comparision.

Point 10: Please provide a limitations section to the discussion. For example, one of the main shortcomings of this study is the confounding effects.

Response 10: Added

Reviewer 2 Report

Thank you very much for that important piece of scientific maritime research. The aspect of BIA offers new aspect of physiological changes in seafarers at sea. A comparison could be made with athletes , astronauts or  hospitalized people. I t would be interesting to compare physical activity on board at the same time. 

Line 21: Does the data give enough information to reflect on the health status ?

Line 24 and discussion: Changes in BMI and BIA reflecting the actual eating and living conditions should not be mixed up with similar changes in the ageing process

Introduction:

Beside the broad reflection on health risks on board an introduction into the BIA method should be given. Results from athletes might interesting as well.

The aspect of physical activity is underrepresented

Line 69-75:   All statements should be solidly sourced

Methods:

Line 116: Why was the body type set to normal for all?

Results:

Table 4: Meaning of the table remains unclear to me. What does the T- and P-values in the table reflect ? the difference between Control and exp-group?

The headline says differences betweenfinal and initial measurement. Then I would expect 2 p-values.

Discussion

The discussion begins with a comparison to the literature. I would recommend to start with a discussion of the values itself.

Line 231: The cited numbers can´t be found in the tables-

Author Response

Response to Reviewer 2 Comments

Point 1: Line 21: Does the data give enough information to reflect on the health status?

Response 1: Yes, we think that changes in body composition is directly related with a changes in a health status, especially if those changes are drastical and un-reverisbile. As we didn’t measured reversiblity of body composition paramaters once the seafarers are back at home we agree that formulation and conclusion is ““pretentious”. We rephrase it to bee more apropriate.

Point 2: Line 24 and discussion: Changes in BMI and BIA reflecting the actual eating and living conditions should not be mixed up with similar changes in the ageing process

Response 2: Corected

Point 3: Beside the broad reflection on health risks on board an introduction into the BIA method should be given. Results from athletes might interesting as well.

Response 3: Added

Point 4: The aspect of physical activity is underrepresented

Response 4: Unfortunately there is no any scientific paper (by our knowledge) in a field of seafaring, or simmiliar profesions. That is issue but also oportunity for the importance of future research in the field.

Point 5: Line 69-75:   All statements should be solidly sourced

Response 5: We added more literature sources

Point 6: Line 116: Why was the body type set to normal for all?

Response 6: Tanita BC-418 device have an option to set body type “normal” or “athlete”. For standarsization of the measurement we set up same settings for all subjects no matter what is their body build and sport preferences. Comparing the “normal” and “athlete” mode for a same subject BIA esitmate lower values of the Body fat and higher values of muscle mass for an “athlete” mode.

We think it is important to emphasize what mode we chose, so the future autors can accurately compare results and methods.

Point 7: Table 4: Meaning of the table remains unclear to me. What does the T- and P-values in the table reflect ? the difference between Control and exp-group?

The headline says differences betweenfinal and initial measurement. Then I would expect 2 p-values.

Response 7: We added paragraph in chapter 2.3 Methods of data analysis to clear the methodological steps for table 4. (now 5.)

“For each measured variables, the differences between the initial and final test of the control and experimental group were calculated and arithmetic means were deter-mined. The differences between initial and final tests of control and experimental group in chronological age, anthropometric characteristics, and body composition measures were determined by using the independent samples T-test.”

Also, we added the new table “Statistical significance of differences in arithmetic means (T-test) of chronological age and anthropometric characteristics of the initial and final test of experimental group” in the paper, folowing the instructions of another reviewer. This table will also make results section more fluid and understandible.

Point 8: The discussion begins with a comparison to the literature. I would recommend to start with a discussion of the values itself.

Response 8: We did it this “old-fashion way” as there is no literature in the field of seafarers. In that section is “all availibele scientific literature” which include any of anthropometric paraemeters of seafarrers. So it is kind of “review paper” within our scientific paper.

Point 9: Line 231: The cited numbers can´t be found in the tables-

Response 9: Those numbers were relative changes in anthropometric parameters form initial and final test. For example, seafarers gained 1.93kg of fat mass what is increase of 10.97%. We changed relative values for the absolute values which are presented in table 5.

Reviewer 3 Report

The manuscript describes one of the main causes of health problems among professional seafarers. Studies have proven that obesity is becoming an upward trend in this professional group. The research was experiential, supported by a statistical analysis of the results obtained. The analyzes were made more detailed thanks to the use of the BIA measurement method. Changes in body composition that occurred in Croatian seafarers during service onboard were analysed.

The title of the publication is consistent with its content. The abstract illustrates the content of the publication well. The keywords are adequate, however the wording seaman, although correct, should be replaced with seafarer as this word is used in the publication. The division into chapters is correct. The selection of current literature is adequate.

The subject matter taken up in the publication is up-to-date, and still insufficiently analysed. The research results are reliable. The selected the group of seafarers who took part in the research programme is adequate. Tools and methods are properly chosen. Furthermore, the authors demonstrated the usefulness of a method not yet utilised in this professional group, allowing for more detailed analyses. Obtained results are correctly interpreted. Their convergence with similar studies is presented. Directions for further research are described too.

Minor editing errors occurred:

Line 191 : no space between population and [19]

Line 201: no space between obese and [20]

Author Response

Response to Reviewer 3 Comments

Point 1: Line 191 : no space between population and [19]

Response 1: Corrected

Point 2: Line 201: no space between obese and [20]

Response 2: Corrected
